# Metabolic Profile Reflects Stages of Fibrosis in Patients with Non-Alcoholic Fatty Liver Disease

**DOI:** 10.3390/ijms24043563

**Published:** 2023-02-10

**Authors:** Nila Jambulingam, Roberta Forlano, Benjamin Preston, Benjamin H. Mullish, Greta Portone, Yama Baheer, Michael Yee, Robert D. Goldin, Mark R. Thursz, Pinelopi Manousou

**Affiliations:** 1Liver Unit, Division of Digestive Diseases, Department of Metabolism, Digestion and Reproduction, Faculty of Medicine, Imperial College London, London W2 1NY, UK; 2Section of Endocrinology and Metabolic Medicine, St Mary’s Hospital, Imperial College NHS Trust, London W2 1NY, UK; 3Department of Cellular Pathology, Faculty of Medicine, Imperial College London, London W2 1NY, UK

**Keywords:** NAFLD, fibrosis, metabolic profile, fast fibrosers

## Abstract

Nonalcoholic fatty liver disease (NAFLD) is a leading cause of chronic liver disease worldwide, with fibrosis stage being the main predictor for clinical outcomes. Here, we present the metabolic profile of NAFLD patients with regards to fibrosis progression. We included all consecutive new referrals for NAFLD services between 2011 and 2019. Demographic, anthropometric and clinical features and noninvasive markers of fibrosis were recorded at baseline and at follow-up. Significant and advanced fibrosis were defined using liver stiffness measurement (LSM) as LSM ≥ 8.1 kPa and LSM ≥ 12.1 kPa, respectively. Cirrhosis was diagnosed either histologically or clinically. Fast progressors of fibrosis were defined as those with delta stiffness ≥ 1.03 kPa/year (25% upper quartile of delta stiffness distribution). Targeted and untargeted metabolic profiles were analysed on fasting serum samples using Proton nuclear magnetic resonance (^1^H NMR). A total of 189 patients were included in the study; 111 (58.7%) underwent liver biopsy. Overall, 11.1% patients were diagnosed with cirrhosis, while 23.8% were classified as fast progressors. A combination of metabolites and lipoproteins could identify the fast fibrosis progressors (AUROC 0.788, 95% CI: 0.703–0.874, *p* < 0.001) and performed better than noninvasive markers. Specific metabolic profiles predict fibrosis progression in patients with nonalcoholic fatty liver disease. Algorithms combining metabolites and lipids could be integrated in the risk-stratification of these patients.

## 1. Introduction

Nonalcoholic Fatty Liver Disease (NAFLD) affects a third of the general population in Western countries [1] and represents the most common cause of abnormal liver function tests. NAFLD includes a spectrum of pathological disorders, from simple steatosis to nonalcoholic steato-hepatitis (NASH) with inflammation and ballooning, and a certain degree of fibrosis up to cirrhosis [2]. The growing prevalence of NAFLD mirrors the epidemic of metabolic syndrome, mainly type-2 diabetes and obesity, to which it is closely associated [1]. Fibrosis stage represents the main predictor of clinical outcomes—liver- and nonliver-related—in this population [3]. As such, developing significant and advanced fibrosis marks a crucial point in the pathogenesis of the disease.

Currently, liver histology represents the gold standard for staging fibrosis in this population [4,5]. However, due to well-identified limitations of liver biopsy, such as the bleeding risk and the cost, it is unfeasible for all the patients to undergo such investigation. A plethora of noninvasive markers, such as ELF score, FIB-4, NAFLD fibrosis score and transient elastography, has therefore been developed over the last few years in an attempt to predict fibrosis stage [6], fast fibrosers and clinical events [7,8,9].

Over the last decade, metabolomic profiling has gained much popularity in the field of translational hepatology. There has been an increasing body of evidence hinting at a possible role of circulating aromatic amino acids as noninvasive markers for NAFLD severity. In a recent study, hepatocellular ballooning and inflammation, assessed by the NASH CRN scoring system, were associated with increased branched chain amino acids and aromatic amino acids, while fibrosis stages could be predicted by a combination of glutamate, serine and glycine [10]. Moreover, plasma branched chain amino acids correlated with NAFLD severity, more pronouncedly in women compared to men [11]. In another study, a combination of glycocholic acid, taurocholic acid, phenylalanine and branched chain amino acids could predict the presence of NASH accurately [12].

In this study, we analysed the metabolic profile of a well-phenotyped cohort of NAFLD patients and investigated its association with fast progressors of fibrosis.

## 2. Results

### 2.1. Study Population

A total of 189 patients were included in the study, with a median follow-up of 72 months (54–106) between their first clinic date to the end of the study and a median time between fibroscan of 15 months (11–22). Median age was 52 (41–60) years. Median BMI was 30.25 (27.7–34.35) kg/m^2^. Non-Hispanic white patients comprised 40.7% (77/189), South Asians 18% (34/189), African/Afro-Caribbeans 6.3% (12/189), White Hispanics 5.8% (11/189), Arabs 5.8% (11/189) and East Asians 4.8% (9/189). In terms of comorbidities, 49.7% (94/189) of the patients had T2DM, 42.9% (81/189) hypertension, 42.9% (81/189) dyslipidaemia and 8.5% (16/189) hypothyroidism (Table 1).

In total, 111 (58.7%) patients underwent a liver biopsy. Overall, 26.7% (50/189) of patients had LSM between 8.1 and 12 kPa, and 18.7% (35/189) of patients had a LSM of more than 12 kPa. A total of 11.1% (21/189) patients were diagnosed with cirrhosis. Based on delta stiffness, 38 (23.8%) patients were classified as fast progressors and 112 (76%) nonfast progressors.

### 2.2. Metabolic Profile in Fast Progressors vs. Nonfast Progressors

When comparing metabolites between fast progressors and nonfast progressors, 38 metabolites were significantly different (Table 2).

On multivariate analysis, only 14 metabolites were significantly associated with fibrosis progression (Table 3). Using binary logistic regression, a formula was generated to predict the progression of fibrosis, based on these metabolites:30.485 × H2TG (mg⁄dL) + 0.706 × H4A1 (mg⁄dL) + 1.586 × H4PL (mg⁄dL) + 1.989 × H4A2 (mg/dL) + 0.063 × V4FC (mg/dL) + 0.719 × IDTG (mg/dL) + 12.94 × VLFC (mg/dL) + 0.103 × V3CH (mg/dL) + 42.376 × Proline (mmol/L) + 3.18 × IDAB (mg/dL) + 0.284 × VLAB (mg/dL) + 44.243 × V3PL (mg/dL) + 0.532 × VLPL (mg/dL) + 0.319 × H2A2 (mg/dL) + 0.339(1)

The metabolic profile generated from the above formula showed the ability to predict fibrosis progression in this population, with an AUROC of 0.788 (95% CI: 0.703–0.874, *p* < 0.001). A cut-off of 0.274 gave a sensitivity of 68% and specificity of 76% (Youden’s J Statistic 0.443). Metabolic profile performed better than other markers for predicting fibrosis progression: ALT (AUROC 0.59, 95% CI: 0.48–0.71, *p* = 0.08), AST (AUROC 0.65, 95% CI: 0.52–0.76, *p* = 0.006), FIB-4 (AUROC 0.5, 95% CI: 0.39–0.61, *p* = 0.9), NAFLD fibrosis score (AUROC 0.56, 95% CI: 0.44–0.67, *p* = 0.26) and LSM at baseline (AUROC 0.43, 95% CI: 0.31–0.55, *p* = 0.2) (Figure 1).

### 2.3. Subgroup with Liver Histology

Among those who underwent biopsy, 15 (14.6%) had stage 0, 22 (21.4%) had stage 1, 19 (18.4%) had stage 2, 35 (34%) had stage 3 and 12 (11.7%) had stage 4 fibrosis. In this cohort, 33 (31.7%) had NASH.

When compared to those without NASH, only phenylalanine was significantly lower in those with NASH (0.07 mmol/L (0.06–0.08) vs. 0.08 mmol/L (0.06–0.09); *p* = 0.048). The AUROC of phenylalanine for predicting the presence of NASH was 0.381 (95% CI 0.269–0.493, *p* = 0.051), which is in keeping with poor diagnostic performance.

When comparing metabolites between fast progressors and nonfast progressors in those who had a biopsy, 37 metabolites were significantly different (Table 4).

On multivariate analysis, only one of these metabolites was significantly associated with fibrosis progression: H4CH, with an OR of 0.818 (95% CI: 0.722–0.927, *p* = 0.002). Using binary logistic regression and including baseline fibrosis stage, a formula was generated to predict the progression of fibrosis:0.818 × H4CH (mg/dL) + 0.793 × Baseline Fibrosis Stage + 13.437(2)

The metabolic profile generated from the above formula showed the ability of prediction of fibrosis progression in this population, with an AUROC of 0.744 (95% CI: 0.618–0.870, *p* = 0.002). A cut-off of 0.203 gave a sensitivity of 85% and specificity of 58% (Youden’s J Statistic 0.433). Metabolic profile performed similar or better than other markers for predicting fibrosis progression: ALT (AUROC 0.69, 95% CI: 0.55–0.84, *p* = 0.016), AST (AUROC 0.75, 95% CI: 0.62–0.89, *p* = 0.001), FIB-4 (AUROC 0.55, 95% CI: 0.40–0.71, *p* = 0.5), NAFLD fibrosis score (AUROC 0.42, 95% CI: 0.26–0.58, *p* = 0.32) and LSM at baseline (AUROC 0.41, 95% CI: 0.25–0.59, *p* = 0.3) (Figure 2).

### 2.4. Metabolic Profile in Cirrhotic vs. Noncirrhotic Patients

In this cohort, 21 (11.1%) patients had cirrhosis. When comparing to the noncirrhotics, 10 metabolites were significantly higher in the cirrhotic group (Table 5). Conversely, 10 different metabolites were significantly lower in the cirrhotic group (Table 5).

On multivariate analysis, only lysine (OR 0.001, 95% CI 0.000002, 0.13, *p* = 0.0008), HDA2 (OR 0.8, 95% CI 0.77, 0.94, *p* = 0.002), H1A2 (OR 1.6, 95% CI 1.2, 2.27, *p* = 0.002) and creatine (OR 6.8 × 10^−15^, 95% CI 8.0 × 10^−29^, 0.58, *p* = 0.046) remained significantly associated with the presence of cirrhosis (Table 6).

Using binary logistic regression, a formula was generated to predict the presence of cirrhosis, combining the metabolites that were significant on multivariate analysis:0.001 × Lysine (mmol/L) + 0.8 × HDA2 (mg/dL) + 1.6 × H1A2 (mg/dL) + 6.827 × 10^−15^ × Creatine (mmol/L) + 27.18(3)

The metabolic profile generated from the above formula showed an excellent prediction of the presence of cirrhosis in this population, with an AUROC of 0.84 (95% CI: 0.752, 0.934, *p* < 0.001). A cut-off of 0.127 gave a sensitivity of 76% and a specificity of 82% (Youden’s J statistic 0.583). Metabolic profile performed similar to the NAFLD fibrosis score and FIB4 but better than other markers for predicting cirrhosis in our cohort: ALT (AUROC 0.33, 95% CI: 0.2–0.46, *p* = 0.02), AST (AUROC 0.44, 95% CI: 0.3–0.59, *p* = 0.48), FIB-4 (AUROC 0.78, 95% CI: 0.66–0.9, *p* < 0.001), NAFLD fibrosis score (AUROC 0.84, 95% CI: 0.74–0.91, *p* < 0.001) and LSM at baseline (AUROC 0.76, 95% CI: 0.62–0.9, *p* < 0.001) (Figure 3).

## 3. Discussion

NAFLD has become the most common cause of liver disease in the Western world and the fastest growing cause for liver transplantation [13]. Despite the big burden of the disease, identifying patients at risk of progression remains a challenge [14]. Of note, the recent advances in metabolomics and lipidomics may provide useful insights into the pathogenesis of the condition as well as new predictive tools for clinical outcomes in this population [15]. In this study, we analysed the metabolic profile of a well-phenotyped group of NAFLD patients from a tertiary care centre. We then evaluated their metabolic profile against fibrosis progression and severity.

Fibrosis stage represents the main predictor of clinical outcomes in patients with NAFLD [3]. Moreover, a faster progression of the liver disease translates into an earlier development of both hepatic and nonhepatic clinical outcomes [16]. As such, identifying those who are at higher risk for fibrosis progression may be clinically important in NAFLD patients. In this cohort, faster progressors presented a peculiar lipid profile, characterised by higher levels of very-low density lipoproteins (VLDL) (VLFC and V3PL) and triglycerides (H2TG) and low HDL (H4CH) (Table 3), which were not observed in cirrhotics (Table 6). Overall, worsening insulin resistance is known to be characterised by elevated VLDL and triglycerides secondary to an impaired hepatic and systemic lipid metabolism [17]. On an opposite trend, sera from patients with cirrhosis were particularly enriched in low-density HDL1 lipoproteins (H1PL, H1CH) and apolipoprotein A2 (HDA2 and TPA2). Specifically, lower levels of VLDL and triglycerides may reflect reduced hepatic synthetic function, greater porto-systemic shunting and relative malnutrition [18]. Moreover, an impaired cholesterol efflux capacity [19], as well as lower lipoprotein scavenger activity [20], may be responsible for elevated HDL and apolipoproteins in those with cirrhosis. There was no difference in terms of statin treatment, suggesting that changes in metabolic profile were due to primary disturbances. Hence, lipid profile may reflect, to some extent, the course of the progression of the liver disease, moving from a phase of florid metabolic dysfunction with fibrosis progression to a less atherogenic profile with established cirrhosis.

Among the metabolites, proline was independently associated with fibrosis progression in this population (Table 3). Proline and its derivate hydroxyproline represent a major player in the collagen synthesis of 30% of the body proteins [21]. Proline also acts as a stabilizer for the helical structure of collagen fibres in the liver [22]. Moreover, previous studies demonstrated that proline uptake increases in early stages of acute steatohepatitis and is proportional to collagenogenesis in animal models [23]. On a similar note, lysine, an essential amino acid mainly catabolised by the liver, was independently associated with the presence of cirrhosis. Previous studies have associated lower lysine levels with collagen disturbances as a result of overexpression of the enzyme lysil oxidases [24]. Under physiological conditions, lysil oxidases deaminate lysine residues for maintaining the structural integrity of the extra-cellular matrix. In pathological conditions such as fibrogenesis, such an enzyme is overexpressed and promotes collagen cross-linking and stabilisation against proteolytic degradation, maintaining hepatic stellate cells in an activated state [25]. Moreover, higher levels of pipecolic acid, one of lysine’s catabolites, were previously described in patients with chronic liver disease and cirrhosis [26]. Taken together, these results suggest a potential role in measuring serum proline and lysine as a biomarker of hepatic collagen turnover in patients with NAFLD. Further prospective studies are required to explore their role as predictors of liver-related events in this population.

Finally, a combination of lipoproteins and metabolites gave an excellent prediction of fibrosis progression (Figure 2) and presence of cirrhosis (Figure 3). With regards to fibrosis progression, metabolic profile performed equal or better than FIB-4, NAFLD fibrosis score and liver functions tests, which are currently used to stratify patients at risk for more severe liver disease in clinical practice [4]. Moreover, while previous studies demonstrated that specific metabolic profiles may distinguish those with simple steatosis from those with NASH [27], this is the first study exploring the association of metabolomics with fibrosis progression and severity. Unfortunately, the small number of events in this population did not allow for an internal validation, while an external cohort was not available to test these findings at the time of the work. Future work should focus validating these results in longitudinal studies as well as external cohorts.

Further studies are required to validate these results in longitudinal studies as well as external cohorts.

In this study, we demonstrated that specific metabolic profile could predict fast fibrosis progression in a cohort of patients with nonalcoholic fatty liver disease. Metabolic profile performed better than traditional noninvasive markers of fibrosis. In an era of precision medicine, algorithms combining metabolites and lipids may provide comprehensive tools to stratify patients with NAFLD. Integrating clinical features and multiomics results may lead to a better understanding of the phenotypes of the patients and may allow for the capture of the complexity of the disease.

## 4. Materials and Methods

### 4.1. Study Population

This study included all consecutive new referrals to the specialist NAFLD clinic at St Mary’s Hospital (Imperial College Healthcare NHS Trust, London, UK) between 2011 and 2019. Exclusion criteria were the use of steatogenic drugs, excess alcohol consumption (defined as alcohol consumption greater than 14 UI per week) as well as other concomitant liver diseases.

Demographic, anthropometric and biochemical data were collected at the time of the baseline fibroscan or at the time of the liver biopsy. Ethnicities were clustered into 6 groups (Table 1). If ethnicity was not specified by the patient, it was classified as Other. All comorbidities, such as hypertension, type-2 diabetes mellitus (T2DM) and hypercholesterolaemia, were recorded. Transient elastography (TE) was performed by an experienced physician after 4 h fasting and allowed for the assessment of liver stiffness measurement (LSM) and controlled attenuation parameter (CAP). A cut-off of LSM ≥ 8.1 kPa was considered to be significant fibrosis, while LSM ≥ 12.1 kPa was considered to be advanced fibrosis [4]. All patients were monitored every 6 months for more than one year, with clinical data documented at subsequent consultation. Liver biopsies were performed when clinically indicated. Liver biopsy specimens were formalin-fixed, paraffin-embedded, stained and scored by an expert liver pathologist as per the NASH CRN scoring system [2]. NASH was defined based on NAS score ≥ 5.

Cirrhosis was diagnosed either histologically or clinically, as a combination of biochemical, imaging and elastographic features. Delta stiffness was calculated as the difference in LSM over a set time period divided by number of months. Patients were divided into fast progressors and nonfast progressors when delta stiffness was more than 1.03 kPa/year or less than 1.03 kPa/year, respectively. This cut off was identified using the top 25% of our cohort, which is in line with previous literature [28].

### 4.2. Metabolic Profile

Fasting serum samples were collected for all patients included in the study. They were then centrifuged and stored at −80 °C in the Imperial Hepatology and Gastroenterology Biobank (Imperial College London, London, UK). Targeted and untargeted metabolomic profiles were carried out using proton nuclear magnetic resonance (^1^H NMR). Overall, values for 27 small metabolites and 112 lipoproteins were obtained for each serum sample, as per published protocol [29].

### 4.3. Statistical Analysis

Distribution of the variables was identified using the Shapiro–Wilk normality test, which suggested a nonparametric distribution of the data, and therefore, nonparametric analyses were applied. Descriptive statistics were presented by the median and interquartile range for continuous variables or number and percentage for categorical variables. Difference between groups was measured using the Mann–Whitney U test and Kruskal–Wallis for continuous variables, while Pearson’s chi-squared was used for categorical variables. A Bonferroni-corrected Dunn’s test was used for pairwise analysis of variables within which there were multiple groups. Significant variables were carried forward to multivariate analysis to identify odds ratio (OR) of the variables associated with clinical outcomes. Binary logistic regression was then used to generate a formula collating the variables which were significantly associated with the clinical outcomes on multivariate analysis. ROC (receiver operating characteristic) curves were used to assess the diagnostic performance of the combination of variables identified with the analysis. Sensitivity, specificity and Youden indexes were estimated for a given cut-off. All tests were two-sided, and a *p*-value 0.05 was considered significant.

Statistical analysis was performed using SPSS (version 28.0; SPSS Inc. Chicago, IBM, Chicago, IL, USA).

### 4.4. Ethics

This study was retrospective and included only fully anonymised data from investigations and assessments performed as per standard of care. As such, ethical approval was not required, as stated by the UK policy framework for health and social care. Liver tissue and plasma were stored at Imperial Hepatology Gastroenterology Biobank, which was fully REC approved by Oxford C Research and Ethics Committee under REC reference 16/SC/0021.

## Figures and Tables

**Figure 1 ijms-24-03563-f001:**
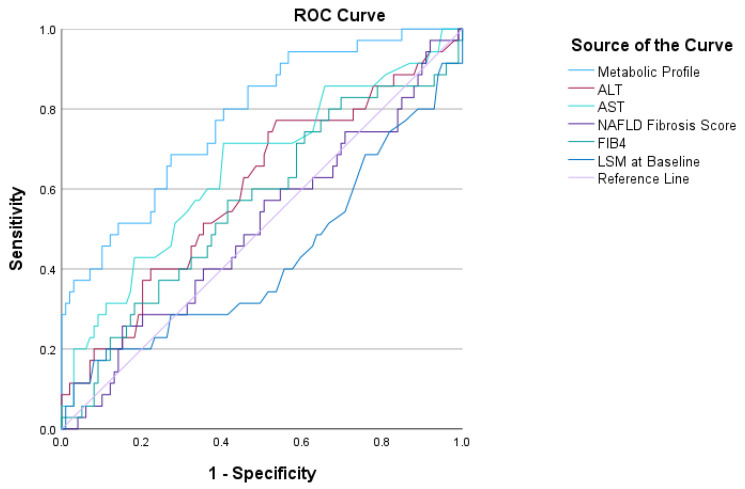
Diagnostic performance of metabolic profile for predicting fibrosis progression compared to ALT, AST, NAFLD fibrosis score, FIB-4 and LSM. Abbreviations: ALT, alanine aminotransferase; AST, aspartate aminotransferase; LSM: liver stiffness measurement.

**Figure 2 ijms-24-03563-f002:**
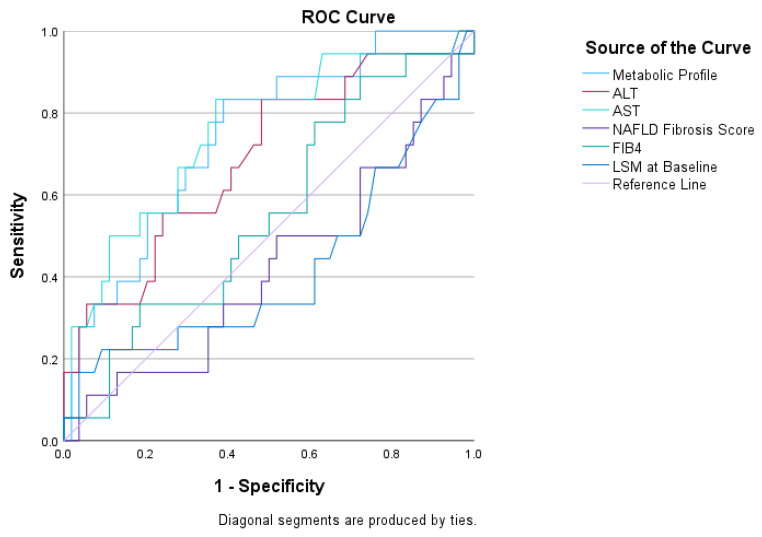
Diagnostic performance of metabolic profile for predicting fibrosis progression compared to ALT, AST, NAFLD fibrosis score, FIB-4 and LSM in patient who had had a liver biopsy. Abbreviations: ALT, alanine aminotransferase; AST, aspartate aminotransferase; LSM: liver stiffness measurement.

**Figure 3 ijms-24-03563-f003:**
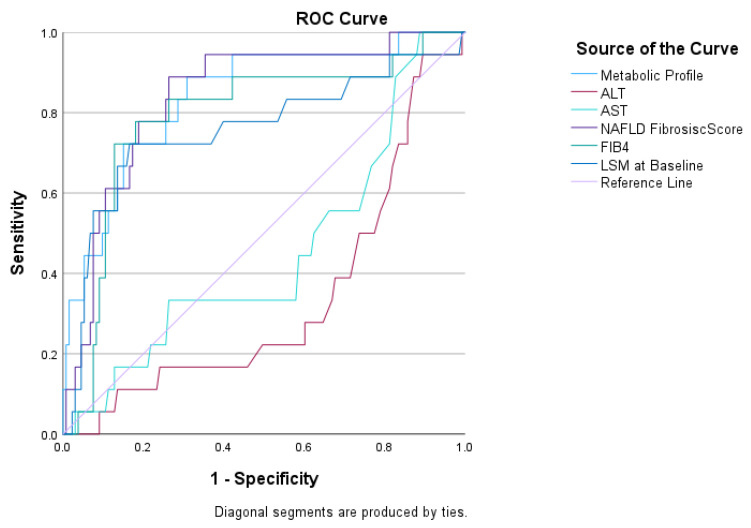
Diagnostic performance of a combination of metabolites to predict the presence of cirrhosis. Abbreviations: ALT, alanine aminotransferase; AST, aspartate aminotransferase.

**Table 1 ijms-24-03563-t001:** Clinical and demographic features of the study population.

	Study Population (n = 189)
**Demographics**	
Male Sex, N (%)	122 (64.6)
Ethnicities, N (%) White Non-Hispanic	77 (40.7)
White Hispanic	11 (5.8)
Arab	11 (5.8)
South Asian	34 (18)
East Asia	9 (4.8)
African/Afro-Caribbean	12 (6.3)
Other	35 (18.5)
Age, median (IQR)	52 (41–60)
Comorbidities	
T2DM, N (%)	94 (49.7)
Dyslipidaemia, N (%)	81 (42.9)
Hypertension, N (%)	81 (42.9)
Hypothyroidism, N (%)	16 (8.5)
Biochemistry	
AST (IU/L), median (IQR)	40 (31–54.5)
ALP (IU/L), median (IQR)	81 (65–104)
GGT (IU/L), median (IQR)	56 (35–105)
PLT (×10^9^/L), median (IQR)	225 (187–271)
HbA1c (mmol/L), median (IQR)	44 (37.25–54)

Abbreviations: T2DM: type-2 diabetes mellitus, AST: Aspartate transaminase, ALP: Alkaline Phosphatase, GGT: Gamma Glutamyl transferase, U/L: units per Litre, L: litre, mmol/L: millimoles per litre.

**Table 2 ijms-24-03563-t002:** Differences in metabolic profile between fast progressors and nonfast progressors.

Metabolite	Fast Progressors (n = 38)	Nonfast Progressors (n = 112)	*p* Value
Glutamic Acid (mmol/L)	0.2	0.15	0.045
Proline (mmol/L)	0.24	0.19	0.035
Valine (mmol/L)	0.31	0.28	0.024
H2A2 (mg/dL)	3.6	3.09	0.042
H3TG (mg/dL)	2.51	2.14	*p* < 0.0001
IDAB (mg/dL)	6.01	5.34	0.039
IDPN (nmol/L)	109.17	97.09	0.039
IDTG (mg/dL)	18.38	13.59	0.024
L3FC (mg/dL)	2.72	3.84	0.034
L4FC (mg/dL)	3.45	3.97	0.042
TPTG (mg/dL)	163.30	140.48	0.022
V1CH (mg/dL)	9.42	7.06	0.013
V1FC (mg/dL)	4.32	3.22	0.014
V1PL (mg/dL)	9.96	7.89	0.04
V1TG (mg/dL)	57.02	45.59	0.044
V2CH (mg/dL)	3.76	2.93	0.018
V2FC (mg/dL)	1.77	1.43	0.014
V2PL (mg/dL)	4.93	3.95	0.03
V2TG (mg/dL)	19.43	16.08	0.031
V3CH (mg/dL)	4.42	3.84	0.034
V3FC (mg/dL)	2.35	1.86	0.033
V3PL (mg/dL)	5.58	4.69	0.04
V4FC (mg/dL)	2.85	2.22	0.019
VLAB (mg/dL)	11.60	9.69	0.04
VLCH (mg/dL)	28.86	22.58	0.016
VLFC (mg/dL)	12.96	10.49	0.033
VLPL (mg/dL)	28.76	24.96	0.042
VLPN (nmol/L)	210.90	176.11	0.04
VLTG (mg/dL)	115.59	92.33	0.036
H1TG (mg/dL)	4.31	3.02	0.002
H2TG (mg/dL)	2.23	1.82	*p* < 0.0001
HDTG (mg/dL)	12.34	10.45	0.001
L5FC (mg/dL)	3.49	4.04	0.021
H4FC (mg/dL)	2.68	3.47	0.006
H4CH (mg/dL)	15.92	18.75	0.002
H4A1 (mg/dL)	64.81	71.96	0.005
H4A2 (mg/dL)	16.46	18.58	0.012
H4PL (mg/dL)	23.28	26.42	0.011

Abbreviations: H2A2, HDL-2 Apolipoprotein A2; H3TG, HDL-3 triglyceride; IDAB, IDL apolipoprotein B100; IDPN, IDL particle number; IDTG, IDL triglyceride; L3FC, LDL-3 free cholesterol; L4FC, LDL-4 free cholesterol; TPTG, total plasma triglycerides; V1CH, VLDL-1 cholesterol; V1FC, VLDL-1 free cholesterol; V1PL, VLDL-1 phospholipid; V1TG, VLDL-1 triglycerides; V2CH, VLDL-2 cholesterol; V2FC, VLDL-2 free cholesterol; V2PL, VLDL-2 phospholipid; V2TG, VLDL-2 triglyceride; V3CH, VLDL-3 cholesterol; V3FC, VLDL-3 free cholesterol; V3PL, VLDL-3 phospholipid; V4FC, VLDL-4 free cholesterol; VLAB, VLDL class apolipoprotein B100; VLCH, VLDL class cholesterol; VLFC, VLDL class free cholesterol; VLPL, VLDL class phospholipid; VLPN, VLDL class particle number; VLTG, VLDL class triglycerides; H1TG, HDL-1 triglyceride; H2TG, HDL-2 Triglyceride; HDTG, HDL triglyceride; L5FC, LDL-5 free cholesterol; H4FC, HDL-4 free cholesterol; H4CH, HDL-4 cholesterol; H4A1, HDL-4 apolipoprotein A1; H4A2, HDL-4 apolipoprotein A2; H4PL, HDL-4 phospholipid.

**Table 3 ijms-24-03563-t003:** Multivariate analysis of metabolites independently associated with fibrosis progression.

Metabolite	OR (95% CI)	Significance
H2TG (mg/dL)	30.48 (4.37–212.57)	<0.001
H4A1 (mg/dL)	0.706 (0.57–0.88)	0.001
H4PL (mg/dL)	1.586 (1.09–2.32)	0.017
H4A2 (mg/dL)	1.989 (1.99–1.18)	0.009
V4FC (mg/dL)	0.063 (0.06–0.007)	0.014
IDTG (mg/dL)	0.719 (0.72–0.57)	0.004
VLFC (mg/dL)	12.94 (1.92–87.15)	0.009
V3CH (mg/dL)	0.103 (0.01–1.02)	0.052
Proline (mmol/L)	42.376 (2.34–767.97)	0.011
IDAB (mg/dL)	3.18 (1.38–7.32)	0.006
VLAB (mg/dL)	0.284 (0.09–0.92)	0.036
V3PL (mg/dL)	44.243 (1.66–1179.2)	0.024
VLPL (mg/dL)	0.532 (0.30–0.95)	0.032
H2A2 (mg/dL)	0.319 (0.12–0.83)	0.019

Abbreviations: H2TG, HDL 2 Triglyceride; H4A1, HDL 4 Apolipoprotein 1; H4PL, HDL 4 Phospholipid; H4A2, HDL 4 Apolipoprotein A2; V4FC, VLDL 4 free cholesterol; IDTG, IDL class triglyceride; VLFC, VLDL free cholesterol; V3CH, VLDL 3 cholesterol; IDAB, IDL class Apolipoprotein B100; VLAB, VLDL class Apolipoprotein B100; V3PL, VLDL 3 Phopholipid; VLPL, VLDL Phospholipid; H2A2, HDL 2 Apolipoprotein A2.

**Table 4 ijms-24-03563-t004:** Differences in metabolic profile between fast progressors and nonfast progressors in patients who had a liver biopsy.

Metabolite	Fast Progressors (n = 22)	Nonfast Progressors (n = 63)	*p* Value
Two Oxoglutaric Acid (mmol/L)	0	0	<0.001
H4CH (mg/dL)	15.545	18.99	0.002
H4PL (mg/dL)	22.28	26.14	0.004
H4FC (mg/dL)	2.52	3.31	0.006
H4A1 (mg/dL)	62.62	71.85	0.009
V1CH (mg/dL)	9.42	6	0.01
H2TG (mg/dL)	2.225	1.82	0.01
V2FC (mg/dL)	1.765	1.21	0.012
V2CH (mg/dL)	3.67	2.81	0.012
VLCH (mg/dL)	28.25	21.46	0.012
V1FC (mg/dL)	4.32	2.69	0.013
H4A2 (mg/dL)	15.9	18.46	0.014
VLFC (mg/dL)	12.9	9.72	0.017
H3TG (mg/dL)	2.53	2.22	0.017
HDTG (mg/dL)	12.52	10.46	0.02
HDFC (mg/dL)	9.28	11.02	0.02
V4FC (mg/dL)	3.135	2.13	0.021
VLPL (mg/dL)	28.34	23.94	0.021
VLPN (mg/dL)	206.75	161.42	0.023
VLAB (mg/dL)	11.37	8.88	0.024
TPA2 (mg/dL)	28.405	30.78	0.025
V3FC (mg/dL)	2.335	1.63	0.027
V2PL (mg/dL)	4.765	3.65	0.028
TPTG (mg/dL)	163.59	127.42	0.028
V2TG (mg/dL)	19	14.99	0.029
V3CH (mg/dL)	4.415	3.54	0.029
H3CH (mg/dL)	8.465	9.21	0.034
VLTG (mg/dL)	112.85	87.7	0.034
HDA1 (mg/dL)	123.85	133.81	0.037
V1PL (mg/dL)	9.275	7.02	0.038
L3FC (mg/dL)	2.69	3.63	0.041
V4CH (mg/dL)	6.18	4.77	0.041
IDTG (mg/dL)	18.005	11.73	0.042
HDA2 (mg/dL)	29.005	31.37	0.044
IDCH (mg/dL)	14.98	11.66	0.045
V3PL (mg/dL)	5.58	4.13	0.049
IDFC (mg/dL)	4.265	3.58	0.049

Abbreviations: H4CH, HDL-4 cholesterol; H4PL, HDL-4 phospholipid; H4FC, HDL-4 free cholesterol; H4A1, HDL-4 apolipoprotein A1; V1CH, VLDL-1 cholesterol; H2TG, HDL-2 triglyceride; V2FC, VLDL-2 free cholesterol; V2CH, VLDL-2 cholesterol; VLCH, VLDL class cholesterol; V1FC, VLDL-1 free cholesterol; H4A2, HDL-4 apolipoprotein A2; VLFC, VLDL class free cholesterol; H3TG, HDL-3 triglyceride; HDTG, HDL class triglyceride; HDFC, HDL class free cholesterol; V4FC, VLDL-4 free cholesterol; VLPL, VLDL class phospholipid; VLPN, VLDL class particle number; VLAB, VLDL class apolipoprotein B100; TPA2, Total Plasma apolipoprotein A2; V3FC, VLDL-3 free cholesterol; V2PL, VLDL-2 phospholipid; TPTG, Total Plasma trigylceride; V2TG, VLDL-2 triglyceride; V3CH, VLDL-3 cholesterol; H3CH, HDL-3 cholesterol; VLTG, VLDL class triglyceride; HDA1, HDL class apolipoprotein A1; V1PL, VLDL-1 phospholipid; L3FC, LDL-3 free cholesterol; V4CH, VLDL-4 cholesterol; IDTG, IDL class triglyceride; HDA2, HDL class apolipoprotein A2; IDCH, IDL class cholesterol; V3PL, VLDL-3 phospholipid; IDFC, IDL class free cholesterol.

**Table 5 ijms-24-03563-t005:** Differences in metabolic profile between cirrhotic and noncirrhotic patients.

Metabolite	Cirrhotics (n = 21)	Noncirrhotics (n = 162)	*p* Value
H1A1 (mg/dL)	33.48	16.95	0.007
H1A2 (mg/dL)	3.49	1.83	0.016
H1CH (mg/dL)	20.43	13.64	0.014
H1PL (mg/dL)	28.42	15.86	0.017
H1FC (mg/dL)	4.29	3.15	0.039
L2TG (mg/dL)	2.73	2.00	0.015
3OHB (mg/dL)	0.09	0.06	0.012
H1TG (mg/dL)	5.3	3.25	0.001
H2TG (mg/dL)	2.48	1.96	0.021
HDTG (mg/dL)	14.57	11.15	0.02
L5FC (mg/dL)	3.35	3.91	0.027
H4FC (mg/dL)	2.4	3.32	0.014
H4CH (mg/dL)	13.35	18.03	0.006
H4A1 (mg/dL)	59.23	70.09	0.014
H4A2 (mg/dL)	15.8	18.46	0.001
H4PL (mg/dL)	19.67	26.09	0.004
HDA2 (mg/dL)	28.09	30.61	0.01
TPA2 (mg/dL)	27.62	30.09	0.012
Creatine (mmol/L)	0.01	0.02	0.036
Lysine (mmol/L)	0.18	0.23	0.001

Abbreviations: H1A1, HDL 1 Apolipoprotein A1; H1A2, HDL 1 Apolipoprotein A2; H1CH, HDL 1 cholesterol; H1PL, HDL 1 Phospholipid; H1FC, HDL 1 free cholesterol; L2TG, LDL 2 triglyceride; 3OHB, 3 hydroxybutyric acid; H1TG, HDL 1 triglyceride; H2TG, HDL 2 triglyceride; HDTG HDL class triglyceride; L5FC, LDL 5 free cholesterol; H4FC, HDL 4 free cholesterol; H4CH, HDL 4 cholesterol; H4A1, HDL 4 Apolipoprotein A1; H4A2, HDL 4 Apolipoprotein A2; H4PL, HDL 4 Phospholipid; HDA2, HDL class Apolipoprotein A2; TPA2, total plasma Apolipoprotein A2.

**Table 6 ijms-24-03563-t006:** Multivariate analysis of the metabolites independently associated with the presence of cirrhosis.

Metabolite	OR (95% CI)	*p*-Value
Lysine (mmol/L)	0.001 (0.000002–0.137)	0.008
HDA2 (mg/dL)	0.856 (0.77–0.94)	0.002
H1A2 (mg/dL)	1.654 (1.2–2.27)	0.002
Creatine (mmol/L)	6.8 (0.008–0.58)	0.046

Abbreviations: OR: odds ratio, 95% CI: 95% confidence interval, HDA2: HDL class Apolipoprotein A2, H1A2: HDL-1 Apolipoprotein A2.

## Data Availability

Data are contained within the article.

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
