# Peer review of "Metabolic Profile Reflects Stages of Fibrosis in Patients with Non-Alcoholic Fatty Liver Disease"

_ijms, 2023, doi:10.3390/ijms24043563_

Round 1

Reviewer 1 Report

Dear authors,

Thank you for submitting this interesting paper on how metabolic profile can predict fibrosis progression in a cohort of patients with biopsy-proven NAFLD.

I have two minor comments.

1. Was a validation cohort considered to test the metabolic scores? If not, this should be clearly stated as limitation of the work.

2. Moreover, considering that metabolic scores here rely mainly on lipoproteins, did the author observe differences in statin use? If so, it would be interesting to expand this point in the paper.

Author Response

Reviewer n.1

Dear authors,

Thank you for submitting this interesting paper on how metabolic profile can predict fibrosis progression in a cohort of patients with biopsy-proven NAFLD.

I have two minor comments.

  1. Was a validation cohort considered to test the metabolic scores? If not, this should be clearly stated as limitation of the work.

            The reviewer makes a very good point. An internal or external validation would have confirmed the reproducibility and applicability of the results. Nevertheless, an external cohort with biopsy and metabolomics result was not available from other centres. Moreover, the number of events was too small to allow for splitting the population into an internal cohort. We have made this point clearer in the discussion – Page 11 line 305-309.

  1. Moreover, considering that metabolic scores here rely mainly on lipoproteins, did the author observe differences in statin use? If so, it would be interesting to expand this point in the paper

            This is an excellent point and we thank the reviewer for raising it. We did not observe any differences in terms of statin use in this population and implemented the discussion. Page 10 line 274-275

Reviewer 2 Report

Congratulations to the authors for this fantastic manuscript that represents an important advance in the diagnosis of the different states of fibrosis in patients with NAFLD.

In addition, they have done so using metabolomics with the analysis of patient serum samples, a sample that is easy to obtain and that can be taken from any patient without causing problems due to the type of sample and dispensing with hospitalizations and interventions in the operating room.

Perhaps in the data capture indicating the ethnicity directly by the patient, it may not be the most correct, but it is valid in most cases, since there is a huge data matrix that the authors have been able to capture in tables and ROC graphs of masterful way.

Surely the data obtained and n number are not sufficient to conclude more precisely on the most relevant data or combination of data to make an accurate diagnosis, mainly so that it can be applied to samples at a clinical level, without the need to perform complete metabolomics. I hope that with the publication of this article, they can get funding to complete this research in the near future, to select the most important parameters to diagnose the stage of fibrosis in patients with NAFLD.

I would only ask the authors to try to incorporate their results in the discussion, since they only make a reference to figure 2 in their entire discussion. Probably with some mention of its results, it may be easier for future readers to understand the interpretation of the results, and may produce a greater impact of the research.

That is why it only requires a minor review to be accepted at the IJMS.

Author Response

            Dear reviewer,

            Thank you very much for your comments. We have signposted Figures and Tables in the discussion to allow for a better integration between results and discussion.